# Efficacy of Endobiliary Radiofrequency Ablation in Preserving Survival, Performance Status and Chemotherapy Eligibility of Patients with Unresectable Distal Cholangiocarcinoma: A Case-Control Study

**DOI:** 10.3390/diagnostics12081804

**Published:** 2022-07-26

**Authors:** Vasile Sandru, Bogdan Silviu Ungureanu, Madalina Stan-Ilie, Ruxandra Oprita, Gheorghe G. Balan, Oana-Mihaela Plotogea, Ecaterina Rinja, Andreea Butuc, Afrodita Panaitescu, Alexandru Constantinescu, Dan Ionut Gheonea, Gabriel Constantinescu

**Affiliations:** 1Department of Gastroenterology, Clinical Emergency Hospital of Bucharest, 105402 Bucharest, Romania; drsandruvasile@gmail.com (V.S.); drmadalina@gmail.com (M.S.-I.); ruxandraa69@gmail.com (R.O.); plotogea.oana@gmail.com (O.-M.P.); ecaterina.rinja@gmail.com (E.R.); butuc_andreea92@yahoo.com (A.B.); afroditapanaitescu@gmail.com (A.P.); gabrielconstantinescu63@gmail.com (G.C.); 2Department of Gastroenterology, University of Medicine and Pharmacy of Craiova, 200349 Craiova, Romania; bogdan.ungureanu@umfcv.ro (B.S.U.); digheonea@gmail.com (D.I.G.); 3Department 5 “Carol Davila”, University of Medicine and Pharmacy, 050474 Bucharest, Romania; 4Gastroenterology Department, Faculty of General Medicine, Grigore T. Popa University of Medicine and Pharmacy Iasi, 700115 Iasi, Romania; 5Department of Gastroenterology, Bucharest University Emergency Hospital, 050098 Bucharest, Romania; alexandruconstantinescu1991@gmail.com

**Keywords:** cholangiocarcinoma, endobiliary frequency ablation, biliary stenting

## Abstract

Background: Cholangiocarcinoma is the most common malignancy of the bile ducts causing intrahepatic, hilar, or distal bile duct obstruction. Most jaundiced patients are diagnosed with unresectable tumors in need for palliative bile duct drainage and chemotherapy. Endobiliary radiofrequency ablation (RFA) is an adjuvant technique that may be applied prior to biliary stenting. The aim of our study was to assess the efficacy of endobiliary RFA prior to stent insertion in patients with unresectable distal cholangiocarcinomas. Methods: Twenty-five patients (eight treated with RFA and stenting and 17 treated with stenting alone) were included in a case-controlled study. We prospectively assessed the impact of RFA on the survival rate, the patient performance status, and the preservation of eligibility for chemotherapy based on the patient laboratory profile. Results: Patients treated with RFA prior to stenting proved to have a significantly longer survival interval (19 vs. 16 months, *p* = 0.04, 95% CI) and significantly better performance status. Moreover, the laboratory profiles of patients treated with RFA has been proven superior in terms of total bilirubin, liver enzymes, and kidney function, thus making patients likely eligible for palliative chemotherapy. Post-ERCP adverse events were scarce in both the study group and the control group. Conclusion: Given the isolated adverse events and the impact on the patient survival, performance, and laboratory profile, RFA can be considered safe and efficient in the management of patients with unresectable distal cholangiocarcinomas.

## 1. Introduction

The most common malignancy of the biliary tree, cholangiocarcinoma, is an epithelial tumor of the bile ducts, anatomically classified as intrahepatic, perihilar, or distal. Intrahepatic cholangiocarcinoma is located proximal to the second-order bile ducts, the perihilar form arises in the right and left hepatic duct or at their junction, while the distal one involves the common bile duct [1]. Although it is considered a relatively rare tumor, representing only 3% of the gastrointestinal malignancies, the overall incidence has gradually increased over the past decades [2]. Unfortunately, the prognosis is usually unfavorable; firstly, because most of the patients are diagnosed with advanced-stage disease, and due to the limited therapeutic options, surgical resection is considered the only potentially curative treatment [3].

Available guidelines suggest that drainage of the biliary tract should be done as soon as possible [4]. While the first considered method was by using a percutaneous approach with plastic drains, currently, the recommended method is to use self-expandable metal stents (SEMS) in an endoscopic setting [5]. However, SEMS have limited patency due to tumor ingrowth or overgrowth and may require additional procedures in time. Local ablative therapies, such as radiofrequency ablation (RFA) [6], photodynamic therapy [7], and cryoablation [8], are considered for several types of cancer because of their capability of inducing cell death within the targeted area.

RFA is a minimally invasive technique that uses high-frequency alternating current inducing resistive heating and coagulative necrosis in order to decrease the size of a tumor. It can be performed by introducing flexible catheters inside the biliary ducts through percutaneous, endoscopic, laparoscopic, and open-surgery approaches. A 1 cm tumor-free or surgical margin is necessary to achieve the RFA goal [9]. RFA was accomplished in a variety of cases of malignant biliary obstruction, such as distal cholangiocarcinoma, pancreatic head carcinoma, and/or gallbladder carcinoma, oncologic efficacy, and survival being comparable with surgical resection in selected patients (e.g., tumor diameter of ≤3 cm) [10].

Currently, RFA is attracting extensive attention, as some reports revealed a longer life stent patency when used for malignant biliary obstruction. However, there is still room for improvement, as a protocol for ablation has not been established yet, the one recommended by the manufacturer’s device is generally used, and also potential adverse events may be encountered. The aim of our study was to analyze RFA outcomes for distal cholangiocarcinoma and to assess the RFA impact on the patient’s survival, performance status, and eligibility for oncologic palliative chemotherapy.

## 2. Materials and Methods

### 2.1. Patients, Procedures, and Study Protocol

A prospective randomized single-center case-controlled study was conducted between 1 January 2019 and 31 December 2021 within the Clinical Emergency Hospital of Bucharest, Romania, including a cohort of 25 patients. The current study was approved by the Ethical Committee of the Clinical Emergency Hospital of Bucharest (1960/28 February 2018). The patients were successively referred by oncological centers after proper workup. We had 8 RFA probes available for the established period of our study. Therefore, we designed a sequenced protocol with a 2:1 randomization rate, as we did not dispose of all patients when the study began. Patients were divided into two groups: there were 8 patients in the study group and 17 patients in the control group. The 2:1 single blind allocation ratio randomization was performed according to the approach sequence, with every RFA patient (study group), and the following 2 patients with distal cholangiocarcinoma were included in the control group.

Inclusion criteria were the age above 18 years old, a positive diagnosis of unresectable distal cholangiocarcinoma with histological confirmation as described below, the availability of complete workup as described below, and the documented informed consent for inclusion. Exclusion criteria were as follows: the presence of any other synchronous malignancy, a positive diagnosis of other types of malignant biliary strictures, the presence of indeterminate strictures, the presence of any counterindication for an ERCP or RFA procedure, the inability to achieve minimal follow-up, the occurrence of any type of surgical intervention within the patients’ history (including palliative surgery), and an estimated survival rate of under 3 months (as evaluated by oncology staff).

The study group underwent endobiliary RFA before bile duct stenting, while the controls underwent palliative ERCP and stent placement alone. All patients underwent complete physical examination and blood panels, along with ultrasonography (US), in order to assess the biliary system; the dilatation of the proximal bile duct was observed in all the cases. Secondly, all included patients underwent a CT scan, as well as a magnetic resonance cholangiopancreatography to precisely identify the location and local extension of the tumor.

The recorded data first included demographics, performance status using Eastern Cooperative Oncology Group (ECOG) scores, and laboratory tests that define eligibility for oncologic therapy as described by Valle et al.: a total bilirubin level of 1.5 times the upper limit of the normal range or less, liver-enzyme levels five times the upper limit of the normal range or less, serum urea and serum creatinine less than 1.5 times the upper limit of the normal range, and a calculated glomerular filtration rate of 45 mL per minute or higher (13). Furthermore, ERCP procedure details and adverse events, type and number of stents placed, and RFA sessions were recorded. Data regarding stricture location, length, diameter, and histology, as well as stent patency were also collected. The minimal follow-up period was 6 months from the first ERCP procedure that included either RFA and plastic stent insertion or plastic stent insertion only. When possible, patients were lifelong followed up. Positive histology or cytology was previously obtained in order to confirm the diagnosis; none of the patients had standard cytology from brushing. Guided biopsies were carried out while performing the ERCP procedure, under a direct view, using Spyglass DS System (Boston Scientific Corporation, Marlborough, MA, United States) during cholangioscopy for control patients. On the other hand, all the patients in the RFA arm had undergone endoscopic ultrasound (EUS) with fine needle aspiration or biopsy (FNA/FNB).

The primary endpoints of the study were to assess the preservation of oncologic eligibility and the performance status of patients. The secondary endpoints included the median survival during the follow-up period, the number of stent changes per patient, and the adverse events rate.

The TJF-Q180V and TJF-Q190V duodenoscopes (Olympus Corporation, Tokyo, Japan) were used for all procedures. Patients underwent the cannulation of the bile duct, and a cholangiogram was performed to assess the site of obstruction. The Habib EndoHBP Bipolar Radiofrequency Catheter (Boston Scientific Corporation, Marlborough, MA, United States) was passed over a guidewire and placed in the region of stricture, 2–3 mm above and under the stricture—the bipolar probe had a diameter of 8 Fr (2.7 mm). Ablation was performed using an Olympus ESG-100 HF generator (Olympus Corporation, Tokyo, Japan) set at level 24 for all the lesions. Each ablation lasted for 120 s followed by a 1 min rest interval. After removing the catheter, the bile duct was cleaned by balloon aspiration to remove residual necrotic tissue and debris (Figure 1). Pre- and post-procedural diameters, as well as the length of the strictures, were measured on the radiologic image. Finally, for both groups, 10 Fr diameter plastic stents were inserted to achieve proper bile duct drainage. Post-ERCP adverse events have been defined as suggested by the European Society for Gastrointestinal Endoscopy (ESGE) and the American Society for Gastrointestinal Endoscopy (ASGE) Guidelines on post-ERCP adverse events [11,12].

### 2.2. Statistical Analysis and Methodology

To evaluate the impact of using RFA interventions, we started from the baseline characteristics of the oncological patient, as they emerge from the monitorization protocol. The following parameters were used, according to the patient’s profile: ECOG scores, the lifetime expectancy, total bilirubin level, the liver enzyme (TGO/TGP), renal function estimated through the levels of serum urea and serum creatinine, and the calculated glomerular filtration rate (eGFR). In case of the GFR, the MDRD methodology was applied considering the gender, the age, and the creatinine level for the white race.

The clinical indicators are described in terms of the mean, the standard deviation, and the 95% confidence interval for the means. In the case of ECOG assessment, percentages and modal values were calculated for the descriptive analysis. As for each patient, the number of interventions, the range of each clinical indicator, and its minimal and maximal levels differed throughout the follow-up. Then, the average min and the average max were calculated as well as the 95% CI for each of them. To compare the groups, the t-test for independent samples was applied for a level of significance alpha = 0.05. Kaplan−Meier survival curves were analyzed, and comparison was made between cumulative survival functions through the Log-rank method.

## 3. Results

Firstly, we assessed the patients’ performance status. The minimal and maximal EGOG scores per patient within the follow-up period were documented (ECOG min and ECOG max, respectively). We identified ECOG performance statuses of 0, 1, 2, and 3 (on a scale ranging from 0 to 5, with lower scores indicating a higher level of functioning). The proportions of ECOG min and ECOG max in the study and control groups are shown in Figure 2. There was an obvious increase in lower scores of ECOG in the study group patients. The comparison of the minimal and maximal ECOG scores between the study and control groups showed a statistically significant difference between average scores: the average value of the minimal EGOG scores was significantly lower in the study group (*p* = 0.04, modal value = 0); also the average value of the maximal ECOG score was significantly lower in the study group (*p* = 0.001 modal value = 1).

Secondly, we evaluated the life expectancy between the two groups (Table 1, Figure 3). At study inclusion, there was an estimated life expectancy of more than three months for all patients. We managed to follow up on survival rates in all patients by registering the time of death. We validated the average survival time in months (the period from the first hospitalization until death) against the life expectancy at inclusion (3 months) in both samples. The 95% confidence interval (CI) for the average survival time alongside with the statistical comparison test of significance against the life expectancy value is shown in Table 1. The comparison between samples also showed that the average survival period was statistically significant longer (19 vs. 16 months) in patients treated with RFA (study group) compared to in patients treated with stenting alone (*p* = 0.04).

Further on, we assessed the impact of the endoscopic intervention on the biological panel of patients regarding the induction or maintenance of patient eligibility for oncologic palliative therapy. The range of total bilirubin (TB) was evaluated in both groups for each patient, considering the level of TB between the first and last stent interventions. The 95% CIs of the minimal−maximal TB limits are shown in Table 2, together with the statistical comparison of the average minimal−maximal values of TB between patient groups. The results showed a significant decrease in TB range in the case of the usage of RFA intervention. There was a statistically significant decrease in both minimal TB (*p* = 0.028) and maximal TB (*p* = 0.017) average values for patients within the study group compared to those in the control group.

The ranges of liver enzyme levels, aspartate aminotransferase (AST), and alanine aminotransferase (ALT) were evaluated in both groups, considering the level of AST/ALT between the first and last stent interventions. The 95% CIs of the minimal and maximal limits and the statistical comparison of the average of the minimum–maximum values of AST/ALT between groups are shown in Table 3. The minimal value of the average AST was statistically significant (*p*-value = 0.035) lower in the study group. However, there was no statistically significant difference between the average values of ALT between groups. Nevertheless, the average of the minimum value of ALT was lower in the study group.

The kidney function was assessed by the follow-up of urea and creatinine levels. The range of serum urea was evaluated in both groups, considering the level of urea between the first and last stent interventions. The 95% CIs of the minimal and maximal values and the statistical comparison of the average minimal and maximal values of urea between groups are shown in Table 4. The minimal urea levels did not significantly differ between samples (*p* = 0.393), but the maximal levels were statistically significantly lower in the case of the study (39.3 mg/dL compared to 53.8 mg/dL; *p* = 0.01). The serum creatinine levels were evaluated in both samples, considering the creatinine levels between the first and last stent interventions. The 95% CIs of the minimal and maximal values and the statistical comparison of the average minimal and maximal creatinine levels between groups are shown in Table 4. Both the average minimal and maximal creatinine levels were significantly lower in case of the study group as opposed to the case of controls: 0.62 mg/dL compared to 1.02 mg/dL (*p* = 0.012) and 0.72 mg/dL compared to 1.44 mg/dL (*p* = 0.008).

Subsequently, we evaluated the eGFRs in both groups. The eGFR scores were calculated through the MDRD method for each patient in both groups, corresponding to the minimal and maximal levels of creatinine. Afterwards, the average minimal and maximal GFR levels were compared between groups. As shown in Table 5, patients treated with RFA and stenting showed significantly higher eGFRs in both minimal creatinine (68.38 ± 31.04 mL/min/1.73 sqm vs. 104.28 ± 11.96 mL/min/1.73 sqm; *p* = 0.022) and maximal creatinine (48.73 ± 20.22 mL/min/1.73 sqm vs. 89.02 ± 11.94 mL/min/1.73 sqm; *p* = 0.001) subgroups compared with in patients treated with stenting alone, thus indicating a better preserved renal function in patients treated with RFA.

Post-ERCP adverse events were exceptional in both the study group and the control group. Interestingly, we detected a post-ERCP bleeding and choledochal-duodenal fistula 72 h after ERCP in one patient with a short (15 mm) and narrow (2 mm) distal cholangiocarcinoma that was managed conservatively with plastic stent on sight (Figure 2). Post-ERCP pancreatitis and cholangitis occurred in less than 5% of patients in both groups, with only one case of post-ERCP cholangitis in the study group. There were no other post-ERCP bleedings or perforations detected.

Due to the low number of patients and the distal localization of the cholangiocarcinomas, a choledocoduodenal fistula was reported after RFA due to the proximity of the malignant stricture and the duodenum (Figure 4). However, because we used pancreatic stents for prophylactic purposes, we did not report any pancreatitis after RFA.

## 4. Discussion

Our case-controlled study showed significantly superior performance levels, longer life expectancies, and better laboratory profiles in patients with unresectable cholangiocarcinoma treated with RFA and stenting (study group) compared to in patients treated with stenting alone (control group). Choosing such endpoints has been determined by the necessity to assure patient eligibility for palliative oncologic therapy as stated by current inclusion standards [13]. Most inoperable distal cholangiocarcinomas are treated with bile duct stenting and palliative chemotherapy. Nevertheless, many patients develop stent occlusion from malignant or hyperplastic tissue ingrowth in stents. At the same time, patients are exposed to recurrent cholangitis, systemic inflammation, organic dysfunction, and multiple hospitalizations.

Traditionally, the supporting data for biliary RFA therapy have been driven by retrospective studies that showed the potential efficiency and safety of RFA when added to standard stenting [14,15]. To date, RFA has gained momentum, and there are several randomized and prospective trials showing that endobiliary RFA prior to stent placement or at the time of stent occlusion might be efficient and safe in prolonging stent patency or even life expectancy [16,17,18,19]. Moreover, a relatively recent meta-analysis showed that when added to stent placement RFA increases both survival and stent patency compared to stent placement alone [20]. Such results are similar to our case-controlled findings. Furthermore, several similarly designed trials also showed the superiority of RFA addon to stenting when compared to bile duct stenting alone [9,21]. However, contrasting results have been reported [22]. Potential bias may arise from the type of stents used in such interventions, as a recent randomized controlled multicenter trial showed that in the case of self-expandable uncovered metallic stents RFA did not prove any positive impact on either stent patency rate or survival [23]. Efficacy might also be influenced by the location of strictures, as RFA has been of greatest interest in hilar lesions where maintaining stent patency and technical success are far more challenging [24,25]. Therefore, further studies comparing different devices, locations, and therapy protocols are warranted.

The draw backs of using RFA described in the literature are as follows: iatrogenic thermal injury that can lead to biliary remodeling or strictures if no biliary stents are inserted, perforation, vessel injury with secondary liver ischemia especially for proximal cholangiocarcinomas (early after the procedure), and/or hemobilia (usually with a 4-week delay) [26].

As shown in both our groups, RFA is a safe procedure, and the reported adverse events are rare [27,28]. Furthermore, RFA technically improved, and newer temperature-controlled RFA devices seem to have a lower risk for excessive depth of injury [29] and thus less frequent bile duct fistulae and bleeding. However, RFA is usually performed in tertiary ERCP centers where adverse event rates would be lower due to local expertise and case selection.

We have chosen a case-controlled design due to the low prevalence and relatively long latency of cholangiocarcinomas. Secondly, as patients were constantly referred to local oncologic therapy, follow-up proved to be difficult in order to prove the outcome of RFA interventions. Thus, a prospective case-controlled design strategy is useful. On the other hand, there are several limitations of our study, mainly including the relatively low number of patients managed in a single center, the selection bias of controls, and the difficulties in following up on comorbidities.

## 5. Conclusions

Our study searched the impact that bile duct RFA has on the patient’s performance status, life expectancy, and laboratory profile when added to standard ERCP with bile duct stenting in patients with distal unresectable cholangiocarcinoma. We showed that when added to plastic stents, RFA increases the performance status and the life expectancy of patients, making them more likely eligible for palliative chemotherapy. It involves a high rate of technical success and only isolated adverse events.

## Figures and Tables

**Figure 1 diagnostics-12-01804-f001:**
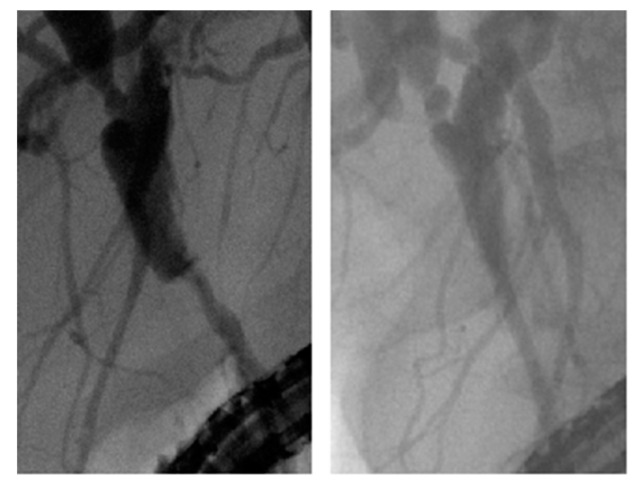
Cholangiocarcinoma before (**left**) and after (**right**) RFA with common bile duct remodeling.

**Figure 2 diagnostics-12-01804-f002:**
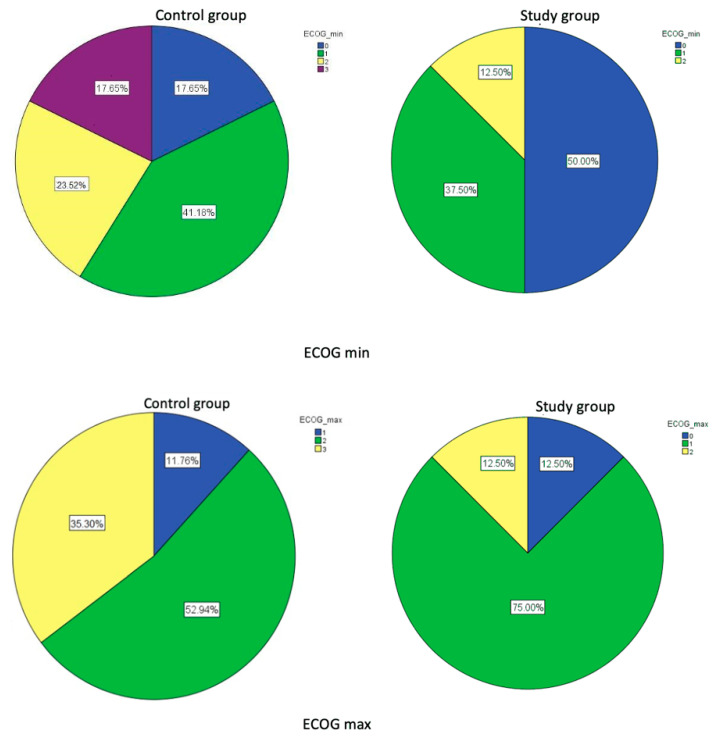
Minimal and maximal EGOC score distributions within the study and control groups.

**Figure 3 diagnostics-12-01804-f003:**
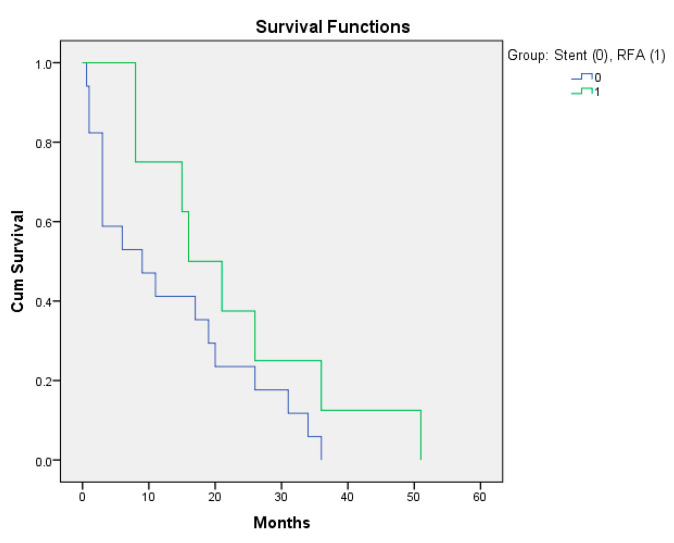
Patients for which the RFA adjuvant method was applied had an increased survival period.

**Figure 4 diagnostics-12-01804-f004:**
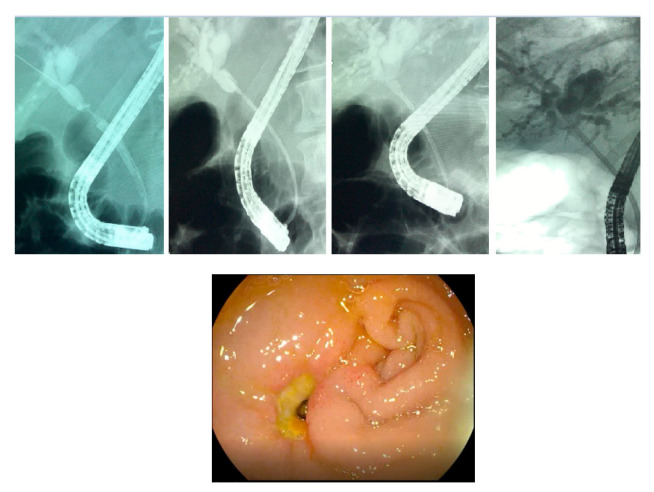
Choledochoduodenal fistula after RFA for cholangiocarcinoma due to proximity of the duodenum.

**Table 1 diagnostics-12-01804-t001:** Survival rates from study inclusion.

Patients	Average Survival (from First Hospitalization until Death)	95% CI for Average Survival Months	*p*-Value of the Comparison Test against the Life Expectancy Value (3 Months)
Control group	16	6.8–25.4 months	0.009
Study group	19	5–33 months	0.035

**Table 2 diagnostics-12-01804-t002:** Variability of the minimal and maximal bilirubin levels between groups.

Patients	Average of the Minimal TB (mg/dL)	95% CI	Average of the Maximal TB (mg/dL)	95% CI
Control group	6.62	2.5–10.7	9.69	6.1–13.2
Study group	3.48	1.1–5.8	5.88	1.9–9.8
*p*	0.028		0.017	

**Table 3 diagnostics-12-01804-t003:** Variability of the minimal and maximal AST and ALT levels between groups.

Patients	Average of the Minimal AST (U/L)	95% CI	Average of the Maximal AST (U/L)	95% CI
Control group	105.53	60.7–150.3	172.33	91.5–253.17
Study group	72.75	29–135.7	105.39	40.6–170.15
*p*	0.035		0.243	
	**ALT(U/L)**		**ALT (U/L)**	
Control group	70.46	43–97.9	175.3	86.3–264.3
Study group	64.12	24.3–103.9	133.12	22.8–244.4
*p*	0.77		0.53	

**Table 4 diagnostics-12-01804-t004:** Variability of kidney function parameters between groups.

Patients	Average of the minimal urea level (mg/dL)	95% CI	Average of the maximal urea level (mg/dL)	95% CI
Control group	39.3	25.12–53.47	53.8	35.7–71.8
Study group	32.5	28.4–36.6	39.3	33.4–45.17
*p*	0.393		0.01	
	**Average of the minimal creatinine level (mg/dL)**	**95% CI**	**Average of the maximal creatinine level (mg/dL)**	**95% CI**
Control group	1.02	0.72–1.32	1.44	0.94–1.94
Study group	0.62	0.6–0.64	0.72	0.59–0.84
*p*	0.012		0.008	

**Table 5 diagnostics-12-01804-t005:** Variability of the average eGFRs (mL/min/1.73 sqm) between groups.

Patients/eGFR	Average eGFR in the Control Group ± St. Dev.	Average eGFR in the Study Group ± St. Dev.	*p*
eGFR/minimal creatinine	68.38 ± 31.04	104.28 ± 11.96	0.022
eGFR/maximal creatinine	48.73 ± 20.22	89.02 ± 11.94	0.001

## Data Availability

The data published in this research are available on request from the first author and the corresponding author.

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
