# Peer review of "Efficacy of Endobiliary Radiofrequency Ablation in Preserving Survival, Performance Status and Chemotherapy Eligibility of Patients with Unresectable Distal Cholangiocarcinoma: A Case-Control Study"

_diagnostics, 2022, doi:10.3390/diagnostics12081804_

Round 1

Reviewer 1 Report

Dear Dr. Sandru and Colleagues,

It was my pleasure to review your manuscript related to the RFA for distal cholangio-ca prior to endo-biliary stent placement. I agree that you have mentioned about the relatively small number of the study subjects. 

Q1. What size Habib probe did you use for RFA?

Q2. Did you place metal CBD stent for any subjects?

Q3. A survival curve would be interesting to add, if possible.

Best wishes

Author Response

Dear Reviewer,

Thank you for your valuable comments which helped us improve the manuscript. We revised the manuscript accordingly. All changes were marked with red colour.

Q1. What size Habib probe did you use for RFA?

Response: We used Habib probe with 8 Fr (2.7 mm) diameter, which was already mentioned in line 124.

Q2. Did you place metal CBD stent for any subjects?

Response: We only inserted plastic stents.

Q3. A survival curve would be interesting to add, if possible.

Response: Thank you for your suggestion. Kaplan-Meier survival curves were added, and comparison was made between cumulative survival functions through Log-rank method. We concluded that the patients for which RFA adjuvant method was applied have a statistically significant increased survival period (p=0.04), measured from first ERCP till death, compared to those treated without RFA. Please see lines 181-187.

Reviewer 2 Report

The authors described "Efficacy of Endobiliary Radiofrequency Ablation in Preserving Survival, Performance Status and Chemotherapy Eligibility of Patients with Unresectable Distal Cholangiocarcinoma: a case-control study." 

They showed that when added to plastic stents, RFA increases the performance status and life expectancy of patients,  making them more likely eligible for palliative chemotherapy. This study should be attractive for potential readers especially in gastrointestinal physicians. However, I have some concerns and suggestions to improve this manuscript.

1. IRB number should be added because this study was a prospective study. Also, how did you divide the two groups? Please add the method in detail. 

2. As shown in this study, RFA was a safe procedure and reported adverse events were rare. How about the adverse events (e.g. infection, hypertrophic scar formation of skin) with RFA in cutaneous lesions? Please describe the draw backs of using RFA.

3. Figures showing typical cases between the groups should be added to make it understandable for the efficacy of RFA.

Author Response

Dear Reviewer,

Thank you for your valuable comments which helped us improve the manuscript. We revised the manuscript accordingly. All changes were marked with red color.

  1. “IRB number should be added because this study was a prospective study. Also, how did you divide the two groups? Please add the method in detail. “

 Response: Thank you for these comments. We added the information that you required with red color in:

Lines 74-78: “A prospective randomized single-center case-controlled study had been conducted between the 1st of January 2019 and the 31st of December 2021 within the Clinical Emergency Hospital of Bucharest, Romania, including a cohort of 25 patients. The current study was approved by the Ethical Committee of the Emergency Clinic Bucharest Hospital, number 1960/28.02.2018.”.

Randomization method has been explained also in the modified version of the manuscript.

Lines 87-90: “Given the relatively low total number of RFA probes which were available during the study period (8 probes) we used a 2:1 single blind allocation ratio randomization. Patients have been divided into two groups: there were 8 patients in the study group and 17 patients in the control group.”

  1. “As shown in this study, RFA was a safe procedure and reported adverse events were rare. How about the adverse events (e.g. infection, hypertrophic scar formation of skin) with RFA in cutaneous lesions? Please describe the draw backs of using RFA.”

Response: Thank you for your suggestion. We added the adverse events (lines 275-278), and also reference number 26 (lines 381-382):

“The draw backs of using RFA described in literature are: iatrogenic thermal injury that can lead to biliary remodeling or strictures if no biliary stents are inserted, perforation, vessel injury with secondary liver ischemia especially for proximal cholangiocarcinomas (early after the procedure) and/or hemobilia (usually with a 4-week delay) [26].“

And lines 239-242:

“Due to the low number of patients and the distal localization of the cholangio-carcinomas, a choledocoduodenal fistula was reported after RFA due to the proximity of the malignant stricture and the duodenum (Figure 4). However, because we used pancreatic stents for prophylactic purposes, we did not report any pancreatitis after RFA. “

  1. “Figures showing typical cases between the groups should be added to make it understandable for the efficacy of RFA.”

Response:  As you suggested, we added figure 1 (lines 136-138) showing Cholangiocarcinoma before and after RFA.

Round 2

Reviewer 2 Report

The authors revised the manuscript. However, I have yet concerns related to study protocol. How did you divide the groups? I understood  a 2:1 single blind allocation ratio randomization, but the method should be mentioned clearly because this study is a prospective study.

Author Response

Dear Reviewer,

Thank you for your valuable comments which helped us improve the manuscript. We revised the manuscript accordingly.

“The authors revised the manuscript. However, I have yet concerns related to study protocol. How did you divide the groups? I understood a 2:1 single blind allocation ratio randomization, but the method should be mentioned clearly because this study is a prospective study.“

 Response: Thank you for your comments. We added the information in the manuscript and marked it with red color:

Lines 74-85: “A prospective randomized single-center case-controlled study had been con-ducted between the 1st of January 2019 and the 31st of December 2021 within the Clinical Emergency Hospital of Bucharest, Romania, including a cohort of 25 patients. The current study was approved by the Ethical Committee of the Clinical Emergency Hospital of Bucharest (1960/28.02.2018). The patients were successively referred by onco-logical centers after proper workup. We had 8 RFA probes available for the established period of our study. Therefore, we designed a sequenced protocol with a 2:1 randomization rate, as we did not dispose of all patients when the study began. Patients were divided into two groups: there were 8 patients in the study group and 17 patients in the control group. The 2:1 single blind allocation ratio randomization was performed according to the approach sequence, with every RFA patient (study group), the following 2 patients with distal cholangiocarcinoma were included in the control group.”